# Changes in Sexual Behavior and Satisfaction and Violent Behavior during COVID-19 Lockdown: Explorative Results from the Italian Cross-Sectional Study of the I-SHARE Multi-Country Project

**DOI:** 10.3390/ijerph21010096

**Published:** 2024-01-15

**Authors:** Filippo Maria Nimbi, Sara Cavagnis, Stefano Eleuteri

**Affiliations:** 1Department of Dynamic and Clinical Psychology and Health Studies, Sapienza University of Rome, 00185 Rome, Italy; filippo.nimbi@uniroma1.it; 2Department of Biomedical and Neuromotor Sciences, University of Bologna, 40126 Bologna, Italy; 3Faculty of Medicine and Psychology, Sapienza University of Rome, 00185 Rome, Italy; stefano.eleuteri@uniroma1.it

**Keywords:** COVID-19 pandemic, sexual and reproductive health, intimate partner violence, Italy

## Abstract

Background: The COVID-19 pandemic has had effects on sexual and reproductive health and intimate partner violence (IPV). This study aims to describe changes in sexual health and IPV in the Italian population both during and after the lockdowns. Methods: This cross-sectional online study was conducted, as part of the I-SHARE multicountry project, between June 2020 and January 2021. Recruitment was carried out through convenience sampling; a total of 329 participants were included in the analysis. A generalized linear model was used to analyze the difference in sexual health and IPV variables before, during, and after the lockdown. Results: Fifty-three percent of the participants reported having sexual problems during the first wave of the pandemic. Sexual satisfaction decreased during the first wave, and then, returned to the pre-lockdown level. While during the lockdown, some activities were reduced (such as kissing, cuddling, and sexual activities with a steady partner), for other activities, no difference was reported (such as masturbation, sexual activities with casual partners, or sexting). Few participants reported having issues accessing HIV testing and contraception. There was no difference in terms of physical and sexual violence, while there was a significant decrease in feeling vulnerable to sexual or physical assault during the first wave. Conclusions: The first wave of the pandemic has had significant effects on sexual health. This should be taken into consideration when preparing for future epidemics and health emergencies.

## 1. Introduction

COVID-19 is a well-known infectious disease that has disrupted the lives of many people all over since the end of 2019. Numerous studies have shown changes in people’s quality of life, habits, and health during and after the COVID-19 pandemic period [1,2].

Some of this evidence focused on exploring possible changes that affected the sexual sphere. The pandemic has clearly conditioned sexual experiences, influencing social relationships around the world due to restrictions and the fear of contracting the infection [3]. For example, many individuals reported that their sexual activity decreased, as physical contact within couples was reduced [4]. Among couples, COVID-19-related stressors predicted a decrease in relational satisfaction and an increase in maladaptive relationship behaviors, such as conflicts [5,6]. According to a systematic review, sexual desire and arousal decreased significantly during the first wave of the pandemic, especially in women, while there was an increase in masturbation and the use of sex toys [7]. Since the beginning of the pandemic, an increase in sexual problems has been found in a range of populations in various countries: both the Female Sexual Function Index and the International Index of Erectile Function were significantly reduced, with a standardized mean difference, respectively of −0.134 and −0.152 [7,8]. During the first wave, women had problems with arousal, orgasm, satisfaction, and pain [9]. Regarding the changes in sexual behavior, an increase in the use of digital means for sexual communication represented one of the most diffused strategies: sexting, cam sex, and the use of pornography have generally increased [4,10].

Considering sexual harassment (and related protection) as an important component of sexual health, there has been concern about COVID-19 exacerbating intimate partner violence (IPV) [11]. Studies on IPV during the first wave offered conflicting results: some studies highlighted an increase in IPV rates during COVID-19 restriction measures [12,13], while others observed a decrease [14,15]. According to a systematic review, the pooled prevalence of IPV during the pandemic was 22% (95% CI 4–40%) [16]. Women were at higher risk of experiencing IPV during the pandemic if unemployed, had low socioeconomic status, a personal or familial COVID-19 diagnosis or mental illness or were exposed to overcrowding [17]. IPV, especially against women, is already a widespread phenomenon in Italy, with a representative prevalence of declared lifetime physical or sexual IPV of 13.6% [18,19]. During the first wave, Italian data on IPV recorded by antiviolence services reported an increase in IPV for 28% of cohabitating women and a decrease for 56% of non-cohabitating ones [20].

Considering the importance of sexual health for general health and well-being [21], there is a need to better understand the impact of social restrictions on individuals’ well-being and their sexuality. To address these and other questions about the impact of COVID-19-related control measures and their effect on sexual behaviors, the I-SHARE consortium (International Sexual Health And Reproductive health survey) was established in early 2020 [22]. It represents a collaborative effort including 30 countries, with the aim of investigating sexual and reproductive health during the COVID-19 pandemic around the world [22]. The present paper aims to present the most relevant I-SHARE results from Italy about sexual and reproductive health changes during and after the first COVID-19 lockdown and social restrictions. Having the opportunity to discuss Italian data is important because it allows us to better understand how different cultures coped with the same globally stressful event. This gives access to a greater level of complexity around COVID-19 effects analysis that includes sociocultural aspects. In addition, Italy was the first Western country to opt for the lockdown measure, as the Italian government proclaimed a total lockdown of all secondary activities from 9 March to 18 May 2020, addressing the novelty of measures restricting personal freedom first [23]. Restrictive measures consisted of maintaining interpersonal distance of at least 1 m in social contacts, hand sanitization at the entrance of any place open to the public, the suspension of educational activities in all schools and universities, the closure of all unnecessary activities and specific restrictions for other public places such as offices and hospitals.

Being in quarantine has been linked to a decline in mental well-being and the emergence of psychological issues like depression, anxiety, insomnia, and post-traumatic symptoms [24]. The loss of routine and limited physical and social interactions may intensify feelings of isolation and loneliness, leading to psychological distress [25]. While existing research primarily focuses on the psychological well-being of adults and healthcare professionals, there is a notable gap in understanding the impact on sexual and reproductive health. This is especially true in the Italian context, where sexual health is still undervalued mainly due to cultural issues. In this sense, the study presented here is very important not only to give a generic perspective on the effects of the first wave of the pandemic, but also to argue the need for improved sexual health policies in Italy.

## 2. Aims

This study aims to provide an overview of the experience of a group of Italians with respect to sexual behavior, access to health services, and IPV.

Specifically, the first objective is to explore any changes in sexual function and satisfaction, the presence of specific difficulties, and access to sexual healthcare when comparing the pre-, during, and post-lockdown periods.

The second objective relates to the possible changes in specific sexual behaviors (such as cuddling, masturbation, sexting, porn, and engaging with steady partners and casual ones) in the pre-, during, and post-lockdown periods.

The third objective is to compare the experience of different types of physical, psychological, and sexual violence in the pre-, during, and post-lockdown periods.

## 3. Materials and Methods

### 3.1. Study Design and Participants

This was a cross-sectional study based on an online survey conducted in Italy between June 2020 and January 2021. This survey is part of a global consortium called I-SHARE, which disseminated the same survey in 30 countries during the first COVID-19 wave. Details on the consortium and the survey methods can be found in the study protocol [22]. Inclusion criteria were being 18 years old or older, living in Italy at the time of the survey, and providing informed consent via an online form. Three hundred forty-five volunteers from the Italian general population participated in the current study. Sixteen responses (4.64%) were excluded from the present study because they represented duplicated, falsified, or incomplete records. The final group resulted in 329 participants (207 women, 110 men, and 12 other genders).

### 3.2. Data Collection

Recruitment was carried out through convenience sampling by the authors: the survey link was shared through the researchers’ network and social media. The link to the questionnaire was accessible via computer, smartphone, and tablet. The questionnaire was adapted from the I-SHARE instrument and translated into Italian. It included sections on sociodemographics, compliance with COVID-19 measures, family and couple relationships, sexual behavior, contraceptive use and barriers to access, access to reproductive healthcare, abortion, sexual violence and IPV, female genital mutilation/cutting and early/forced marriage, HIV/STI testing and treatment, mental health, and food insecurity. In the Italian version, in addition to the I-SHARE survey, in some sections (such as sexual behavior or sexual violence), participants were asked about their experiences before, during, and after the lockdown. The survey consisted of 208 questions, most of which were optional; it took around 25 min to complete. The instrument with coded variables can be found in the Appendix A.

### 3.3. Data Analysis

Analysis was carried out using SPSS v27.0. First, descriptive analyses of the sociodemographic variables were conducted, followed by analyses of changes in reported behavior (described as frequencies and percentages). For the second and third objectives of the study, a generalized linear model for repeated measures was used to analyze the difference in sexual satisfaction, sexual problem frequency, and experiences of sexual violence and IPV before, during, and after the lockdown.

## 4. Results

The sociodemographic characteristics of the participants are reported in Table 1. Participants’ ages ranged between 19 and 79 years old, with a mean age of 32.26 ± 10.27 (Q_3_–Q_1_: 25–38). The group was composed of 2/3 participants assigned female at birth, with most of the participants identifying as women (62.92%), heterosexual (71.17%), and reporting that they were in a romantic relationship (68.09%). Most of them did not have children (86.32%), had a medium to high level of education (98.78%), were students or employed (68.58%), and were living predominantly in large urban areas or small residential areas (93.62%). The socioeconomic status before the COVID-19 pandemic was medium to medium–high for most of the participants (66.56%), with 34.04% declaring a decrease in their income status in the first wave. Most of the participants declared that they were white/Caucasian (97.57%), Christian catholic (40.73%), or had no religion (51.37%).

More than 95% of the group reported having a sexual experience in life, referred to as “any kind of experience that the person feels sexually arousing (such as kissing, touching, intercourse, masturbation, watching sexually explicit images, or any other form of sex)”. In line with the first objective, Table 2 describes some sexual experiences Italian participants had during COVID-19 lockdown/social distancing measures. More than half of the group (52.58%) reported having had sexual problems during the lockdown. The most reported sexual claims were around desire: 30.40% of participants reported having a decrease in sexual desire, while 6.08% reported an increase. Other areas affected were orgasm (3.65%) and pain during sexual penetration (3.04%). Regarding the associated distress, 13.37% of the participants reported significant distress, and 33.43% reported medium/mild distress. Regarding contraceptive use during social distancing measures, 36.78% were using some kind of contraceptive at the time of the survey, and 3.34% of participants found it more difficult to access condoms during lockdowns. Regarding testing for HIV or other sexually transmitted infections (STIs), during the lockdowns, 6.38% of the participants wanted to be tested, but 3.95% were stopped or hindered from accessing a test because of the COVID-19 situation.

Table 3 reports the output of a generalized linear model for repeated measures for the analysis of variance in the difference in satisfaction with one’s sexual life and the frequency of sexual problems in three distinct periods: before COVID-19 (about 3 months before), during the COVID-19 lockdown, and after the lockdown (with the persistence of some social restrictions). In the first generalized linear model, satisfaction was set as a dependent variable, and time was set as an independent one. In the second generalized linear model, the presence of sexual problems was set as a dependent variable, and the time was set as an independent one. Both with regard to satisfaction (F = 175.463; *p* < 0.001) and the presence of sexual problems (F = 50.494; *p* < 0.001), there was a significant difference in the three periods, with a peak of dissatisfaction/increase in sexual problems during the lockdown, and then, a return to the pre-lockdown level (Figure 1 for sexual satisfaction and Figure 2 for sexual problems).

Regarding the frequency of sexual experiences before, during, and after COVID-19, different actions were evaluated, ranging from cuddling to online sexual activities (Table 4). Comparing before and during the COVID-19 lockdown, the majority of participants reported a reduction in activities such as hugging, kissing, holding hands or cuddling, and sexual activities with their steady partners. At the same time, no difference was reported by most of the participants in contraceptive use, masturbation, sexual activities with casual partners, sexting, sex in exchange for money and other goods, porn consumption, and other online sexual activities before and during the COVID-19 lockdown.

Upon comparing during and after the COVID-19 lockdown, most participants self-reported an increase in activities such as cuddling and sexual activities with their steady partners. No difference was reported by the majority of the participants in contraceptive use, masturbation, sexual activities with casual partners, sexting, sex in exchange for money and other goods, porn consumption, and other online sexual activities during and after the COVID-19 lockdown.

In line with the third objective, the differences in experiences of sexual violence before, during, and after the lockdown were explored. Table 5 reports the output of a generalized linear model for repeated measures for the analysis of variance in the difference in different kinds of violence before COVID-19 (about 3 months before), during the COVID-19 lockdown, and after the lockdown (with the persistence of some social restrictions). In the generalized linear models, violence-related variables were set as a dependent variable, and time was set as an independent one. Regarding feeling vulnerable to sexual harassment or sexual, physical, or emotional assault, the evaluation was made only before and during the COVID-19 lockdown, highlighting a significant decrease in feelings of vulnerability during the lockdown, as reported in Figure 3 (F = 37.574; *p* < 0.001). No differences were highlighted for family contact restrictions by the partner, insults, and money restrictions in the three periods assessed. No differences were found for physical and sexual violence, although there was an increase in participants declaring “Not applicable to my situation” during the COVID-19 lockdown.

## 5. Discussion

This exploratory cross-sectional study attempted to give a panoramic view of how the lockdown and other COVID-19 restriction measures affected different aspects of sexuality in the short term in Italy. Although the sample is not representative of the Italian population, the sociodemographic data describe a fairly diverse group, in which about 34% experienced a socioeconomic decline during the COVID-19 period. This element may let us consider how the first wave of the pandemic has significantly affected people’s quality of life, resources, and well-being, changing daily priorities in the short and medium term [26,27].

More than 95% of the sample reports being sexually active at the time of participation. During COVID-19 restriction measures, more than 53% of participants reported having some type of sexual difficulty. The most reported sexual claims were in the realm of desire: 30.4% of participants reported having a decrease in sexual desire, while 6.1% reported an increase. In addition to desire, difficulties in the areas of orgasm and pain during intercourse or practices followed. These data are consistent with the results found in another Italian study [28] and from other countries [9]. Cross-referencing this with sexual distress, only 13.4% of participants rated the distress related to their sexual problem as significant versus 33.4% who rated it as a problem of small/medium significance. The general psychosocial stress due to the first wave (and the related restrictions) may have played a significant role in the onset and maintenance of sexual symptoms [29]. In this sense, we can hypothesize how the sexual difficulties encountered by some participants may present themselves in a reactive form to acute stress, such as having to cope with a pathogen that, at the time, was unknown and restrictive social measures never faced before. This connection would seem to be reinforced by the fact that the incidence of sexual problems and dissatisfaction significantly decreased after the end of the lockdown, showing results similar to pre-pandemic. It seems to be no coincidence that the area of desire appears to be the one most affected by the first COVID-19 wave. In fact, desire is a highly subjective experience that is directly affected by psychological and emotional components, like many sources of stress [30,31].

Regarding access to sexual health tools and services, our results showed that a few participants found more difficulties having access to condoms (3.3%) and expressed their desire to be HIV tested (less than 7%), and 3.9% had problems in accessing it. In the global I-SHARE data, 38.2% of participants who needed HIV/STI testing reported that COVID-19 measures hindered them from accessing prevention care [14]. A similar trend with smaller differences was evident in the difficulties in finding condoms: in the global I-SHARE data, COVID-19 measures impeded access to condoms in 8.7% of people. It is possible that taking care of sexual health with STI testing and contraceptives was not a priority during the lockdown for most of the participants in this study. Many participants were in monogamous relationships and were not living with their partner during lockdown. Moreover, casual sexual encounters were limited by social restriction measures. In this sense, for some, the perception of having to protect themselves from unwanted pregnancies or STIs may have been influenced, putting on the back burner the need to use contraceptives and test for STIs. The health focus (both by institutions and the public) was mainly shifted to COVID-19 infection in early 2020, overshadowing the importance of sexual and reproductive health. Moreover, it should be considered that during the lockdowns, it was possible to always have access to condoms in pharmacies and supermarkets: this could also contribute to the fact that just a few people perceived impeded access to condoms in our country.

Considering specific sexual activities before, during, and after the COVID-19 lockdown, most participants reported a decrease in cuddling and sexual activities with their steady partners during the lockdown compared to the 3 months before the pandemic. This result might be the expression of those participants who were not cohabiting with their partners during the social restrictions. In fact, the rates increased after the end of the lockdown, consistent with the reunion with their partners. No difference was reported by the majority of participants in contraceptive use, masturbation, sexual activities with casual partners, sexting, sex in exchange for money and other goods, porn consumption, and other online sexual activities during and after the COVID-19 lockdown. These results regarding behavioral changes during the lockdown are consistent with the ones reported in Germany [32] and England [33]. More specific differences were expected for sexual activity changes and sociodemographic variables such as relationship type, gender, age, living together, and others. At present, it is not possible to go into detail due to the small number of participants and the structure of the I-SHARE questionnaire. This limitation did not allow for a more detailed statistical analysis of the data, leaving unanswered these questions. The international literature has shown an increase in masturbatory activity, pornography consumption, and use of remote sexual activities such as sexting and cam sex [34,35]. It is possible that these types of activities, which were already widespread before the COVID-19 restrictions, increased, especially in those people who were already familiar with the practice and in people who were motivated to engage in sexual activities as a reactive coping strategy to social stress. Future research should try to investigate the specificity of changes in sexual activities, perhaps using larger databases collected during the pandemic period such as the I-SHARE multi-country survey.

Regarding different forms of sex-related violence explored by the I-SHARE protocol, in our study, no significant differences were found in the three periods assessed. The only exception was for the sense of vulnerability felt due to harassment and assault by not cohabitant people. Fewer people felt vulnerable during the restrictions, probably due to the sense of isolation and protection experienced in the lockdown, where most people lived in their homes, going out only for primary reasons (shopping for basic necessities, health issues, heading to the workplace in some cases). Living in their homes may have diminished the feeling of vulnerability, while also limiting the actual risk of being raped by strangers. This is consistent with the “stranger danger” effect, which gives the idea that being in the house with people you know could diminish the risk of violence [36]. These results are also in line with the findings of the global I-SHARE study, according to which there was a modest decrease in sexual and physical partner violence during COVID-19 measures compared to the pre-COVID period [14,37].

While there is a lack of data on what happened after the end of the various lockdowns and first-wave stress in general, new studies should focus on the medium- and long-term effects of the global social stress that the pandemic had (and continues to have) on sexual and reproductive health.

The current study acknowledges certain limitations. Firstly, participants were recruited using a “snowball” technique and social media advertising, making it challenging to generalize the findings to the Italian population, despite the diverse range of participants. Secondly, the research design of this study is cross-sectional, although it asks about the perception of participants at three different times (pre-, during, and post- lockdown). The I-SHARE measure was specifically developed and adapted for the Italian language in early 2020. While initial psychometric analyses have been undertaken, further research is needed to explore the validity and reliability of this measure. Lastly, the study relied on self-reported measures, potentially allowing respondents to falsify their responses. Another potential bias is that it was not possible to control whether participants traveled during restriction measures in other countries, although February–June 2020 was not a period when people could travel freely.

Future studies should consider how sexual and reproductive health has evolved not only since the emergency restriction measures, but especially in the long run, after the WHO’s declaration of the end of the pandemic in June 2023. This temporal overview could be extremely useful in understanding the complexity of the pandemic’s effect and planning preventive sexual health measures for possible future emergencies.

## 6. Conclusions

This study adds to the existing evidence of sexual behavioral changes during the first wave of the COVID-19 pandemic. There are now a lot of data about the impact of lockdown on sexuality, which has, on one hand, worsened sexual health problems, and on the other hand, highlighted new possibilities for engaging in one’s sexual life, such as self-eroticism and digital sex. These changes should be taken into consideration when planning responses to health emergencies and epidemics. This study can specifically provide some intervention and prevention directions related to social policy and clinical practice. For example, it is suggested by this study that in times of stress, sexuality turns out to be one of the strongly impacted elements. Because sexuality also plays an important role in relationships and stress relief, it is important to develop awareness campaigns that enhance the role of sexual health and increase access to specialized services that can indicate, clarify, and intervene where needed. Sexuality is certainly not the central element when it comes to airborne infectious pathogen emergencies, but it can be a resource for what concerns social restraint measures and a person’s psychological and emotional health.

## Figures and Tables

**Figure 1 ijerph-21-00096-f001:**
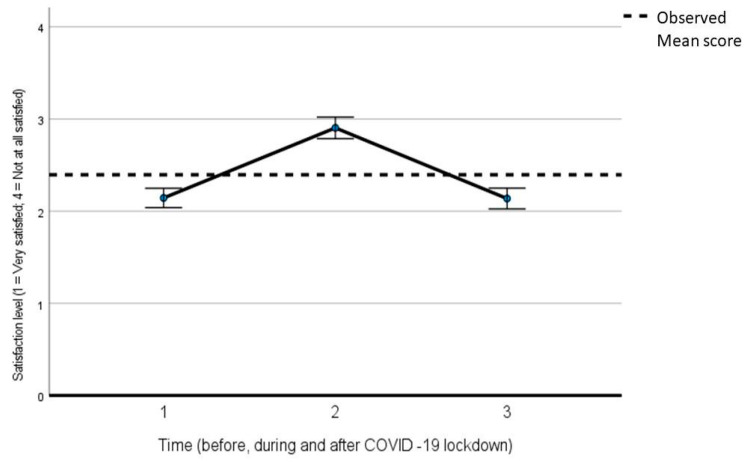
Generalized linear model for repeated measures for the analysis of variance in the difference in sexual satisfaction before, during, and after COVID-19 lockdown.

**Figure 2 ijerph-21-00096-f002:**
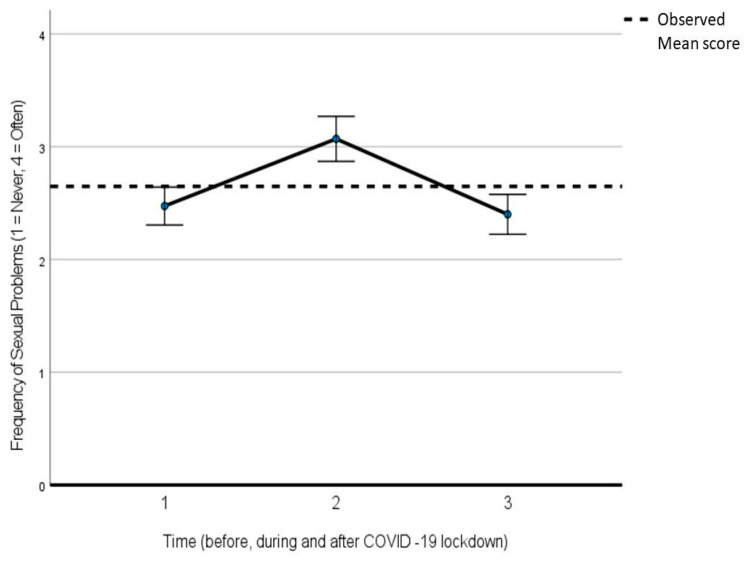
Generalized linear model for repeated measures for the analysis of variance in the difference in sexual problems before, during, and after COVID-19 lockdown.

**Figure 3 ijerph-21-00096-f003:**
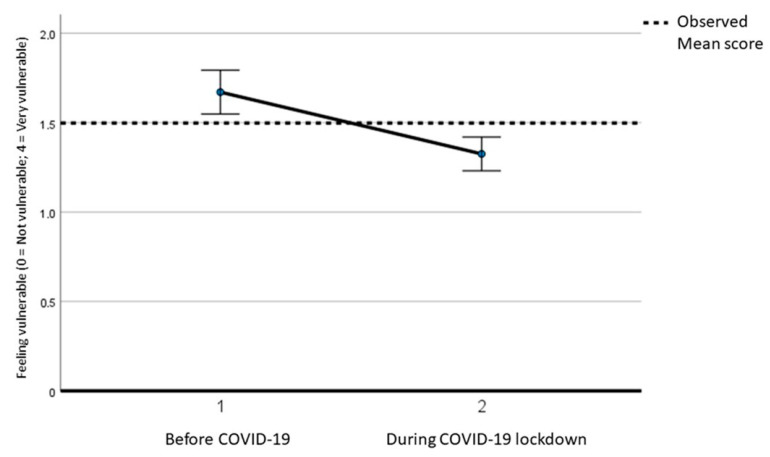
Generalized linear model for repeated measures for the analysis of variance in the difference in feeling vulnerable to sexual harassment or sexual, physical, or emotional assault before and during the COVID-19 lockdown.

**Table 1 ijerph-21-00096-t001:** Sociodemographic variable descriptions.

Variables		Participants (*n* = 329)
		M ± ds (Min–Max)
Age		32.26 ± 10.27 (19–79) Q_3_–Q_1_: 25–38
		*n* (%)
Sex Assigned at Birth	Female	216 (65.65)
	Male	113 (34.35)
Gender	Female	207 (62.92)
	Male	110 (33.43)
	Both	10 (3.04)
	Neither of the two	2 (0.61)
Sexual Orientation	Heterosexual	237 (71.17)
	Bisexual	31 (9.31)
	Homosexual	50 (15.02)
	Pansexual	11 (3.34)
	Asexual	4 (1.22)
Relationship Status	Single	105 (31.91)
	In a relationship	224 (68.09)
Number of Children	0	284 (86.32)
	1	27 (8.21)
	2	15 (4.56)
	3+	3 (0.91)
Education Level	Middle School	4 (1.22)
	High School	48 (14.59)
	University and Postgrad courses	277 (84.19)
Work Status	Employed	120 (36.47)
	Freelance	40 (12.16)
	Unemployed	22 (6.69)
	Fragmentary work	37 (11.25)
	Retired	2 (0.61)
	Student	105 (32.11)
Socio-economic status (1 year before COVID-19 pandemic)	Low	17 (5.17)
	Low/Middle	62 (18.84)
	Middle	130 (39.51)
	Middle/High	89 (27.05)
	High	12 (3.65)
	Prefer not to say	19 (5.78)
Income status changes in first COVID-19 wave	Decrease	112 (34.04)
	No change	205 (62.31)
	Increase	12 (3.65)
Residence	Metropolis	102 (31)
	City	108 (32.83)
	Suburbs	15 (4.56)
	Small town/village	98 (29.79)
	Rural/remote area	6 (1.82)
Ethnicity	White/Caucasian	321 (97.57)
	Latin-American/Hispanic	4 (1.22)
	Black/African/Afroamerican	2 (0.61)
	Prefer not to say	2 (0.61)

**Table 2 ijerph-21-00096-t002:** Descriptives of sexual experience during COVID-19 lockdown in Italy.

Variables		Participants (*n* = 329)
		*n* (%)
Have you ever had a sexual experience?	No	16 (4.86)
By “sexual experience” we mean any kind of experience that you felt was sexually arousing. It could be kissing, touching, intercourse, masturbation, watching sexually explicit images, or any other form of sex	Yes	313 (95.14)
Sexual problems	Hypoactive/low sexual desire	100 (30.40)
	Hyperactive/increase in sexual desire	20 (6.08)
	Low sexual excitation/arousal	9 (2.74)
	Persistent and not wanted genital arousal (in absence of sexual interest)	4 (1.22)
	Erection difficulties	7 (2.13)
	Orgasm difficulties/delayed/impossible ejaculation	12 (3.65)
	Premature ejaculation	6 (1.82)
	Pain during intercourses	10 (3.04)
	Difficulties in allowing vaginal/anal penetration	3 (0.91)
	No sexual problems	156 (47.42)
	Other problems (no specified)	2 (0.61)
Sexual Distress	No problem	176 (53.50)
Did this difficulty represent a problem for you?	A little problem	56 (17.02)
	A problem	54 (16.41)
	A significant problem	29 (8.81)
	A very important problem	15 (4.56)
Did the COVID-19 social distancing measures make it more difficult to access condoms?	No	128 (38.91)
	Yes	11 (3.34)
	Not applicable to my experience—I do not normally use condoms	138 (41.95)
	Missing	52 (15.81)
Are you or your partner currently doing something to avoid or delay a pregnancy, including condoms, contraceptive methods, traditional methods, etc.?	No	39 (11.85)
	Yes, always	99 (30.09)
	Yes, most of the times	17 (5.17)
	Yes, sometimes	5 (1.52)
	Missing	196 (59.57)
During the COVID-19 social distancing measures have you wanted a test for HIV or another sexually transmitted infection?	No	276 (83.89)
	Yes	21 (6.38)
	Missing	32 (9.73)
Has the COVID-19 situation stopped or hindered you from accessing a test for HIV or another sexually transmitted infection?	No	131 (39.82)
	Yes	13 (3.95)
	Missing	185 (56.23)

**Table 3 ijerph-21-00096-t003:** Sexual satisfaction and sexual problems before, during, and after COVID-19 lockdown.

Variable		Pre-COVID-19 (about 3 Months before)	During COVID-19 Lockdown	After COVID-19 Lockdown	F _(1,312)_	Sign
Satisfaction with Sexual Life	Very satisfied	88 (26.75)	37 (11.25)	100 (30.40)	175.463	<0.001
	Quite satisfied	124 (37.59)	77 (23.40)	112 (34.04)		
	Not very satisfied	69 (20.97)	78 (23.71)	59 (17.93)		
	Not at all satisfied	32 (9.73)	121 (36.78)	42 (12.77)		
	Missing	16 (4.86)	16 (4.86)	16 (4.86)		
Frequency of Sexual problems	Never	105 (31.91)	88 (26.75)	123 (37.39)	50.494	<0.001
	Once	29 (8.81)	14 (4.26)	19 (5.78)		
	Sometimes	86 (26.14)	54 (16.41)	68 (20.67)		
	Often	18 (5.47)	24 (7.29)	23 (6.99)		
	Not applicable to my situation	41 (12.46)	93 (28.27)	43 (13.07)		
	Missing	50 (15.20)	56 (17.02)	53 (16.11)		

**Table 4 ijerph-21-00096-t004:** Frequency of sexual activities before, during, and after COVID-19.

	Pre-COVID-19 (about 3 Months before)		During COVID-19 Lockdown (Compared to before COVID-19)		After COVID-19 Lockdown (Compared to during Lockdown)	
Hugged, kissed, held hands with, or cuddled with your steady partner	Never	12 (3.65)	Much less	86 (26.14)	Much less	13 (3.95)
	Once a month or less	8 (2.43)	A bit less	32 (9.73)	A bit less	30 (9.12)
	2–4 times per month	27 (8.21)	No difference from before	58 (17.63)	No difference from before	80 (24.32)
	2–3 times per week	50 (15.20)	A bit more	29 (8.81)	A bit more	49 (14.89)
	4 or more times per week	140 (42.55)	Much more	23 (6.99)	Much more	62 (18.84)
	Missing	92 (27.96)	Missing	101 (30.70)	Missing	95 (28.88)
Engaged in sexual activities with your steady partner	Never	17 (5.17)	Much less	94 (28.57)	Much less	20 (6.08)
	Once a month or less	15 (4.56)	A bit less	31 (9.42)	A bit less	29 (8.81)
	2–4 times per month	97 (29.48)	No difference from before	75 (22.80)	No difference from before	82 (24.92)
	2–3 times per week	78 (23.71)	A bit more	19 (5.78)	A bit more	59 (17.93)
	4 or more times per week	33 (10.03)	Much more	8 (2.43)	Much more	44 (13.37)
	Missing	89 (27.05)	Missing	102 (31)	Missing	95 (28.88)
Used condoms/contraceptives when you had sex with your steady partner	Never	104 (31.61)	Much less	26 (7.90)	Much less	20 (6.08)
	Rarely	9 (2.74)	A bit less	5 (1.52)	A bit less	3 (0.91)
	Sometimes	17 (5.17)	No difference from before	175 (53.19)	No difference from before	186 (56.53)
	Most of the time	17 (5.17)	A bit more	2 (0.61)	A bit more	7 (2.13)
	Always	86 (26.14)	Much more	2 (0.61)	Much more	10 (3.04)
	Missing	96 (29.18)	Missing	119 (36.17)	Missing	103 (31.31)
Masturbation	Never	52 (15.81)	Much less	26 (7.90)	Much less	33 (10.03)
	Once a month or less	45 (13.68)	A bit less	43 (13.07)	A bit less	64 (19.45)
	2–4 times per month	81 (24.62)	No difference from before	128 (38.91)	No difference from before	129 (39.21)
	2–3 times per week	85 (25.84)	A bit more	67 (20.36)	A bit more	54 (16.41)
	4 or more times per week	33 (10.03)	Much more	31 (9.42)	Much more	14 (4.26)
	Missing	33 (10.03)	Missing	34 (10.33)	Missing	35 (10.64)
Engaged in sexual activities with casual partners	Never	231 (70.21)	Much less	55 (16.72)	Much less	19 (5.78)
	Once a month or less	26 (7.90)	A bit less	4 (1.22)	A bit less	4 (1.22)
	2–4 times per month	27 (8.21)	No difference from before	212 (64.44)	No difference from before	211 (64.13)
	2–3 times per week	7 (2.13)	A bit more	2 (0.61)	A bit more	35 (10.64)
	4 or more times per week	0	Much more	3 (0.91)	Much more	8 (2.43)
	Missing	38 (11.55)	Missing	53 (16.11)	Missing	52 (15.81)
Used condoms/contraceptives when you had sex with casual partners	Never	42 (12.77)	Much less	10 (3.04)	Much less	10 (3.04)
	Rarely	11 (3.34)	A bit less	0	A bit less	3 (0.91)
	Sometimes	11 (3.34)	No difference from before	91 (27.66)	No difference from before	91 (27.66)
	Most of the time	20 (6.08)	A bit more	2 (0.61)	A bit more	3 (0.91)
	Always	36 (10.94)	Much more	1 (0.30)	Much more	4 (1.22)
	Missing	209 (63.53)	Missing	225 (68.39)	Missing	218 (66.26)
Sent or received naked/semi-naked pictures or videos	Never	170 (51.67)	Much less	19 (5.78)	Much less	43 (13.07)
	Once a month or less	65 (19.76)	A bit less	10 (3.04)	A bit less	34 (10.33)
	2–4 times per month	30 (9.12)	No difference from before	169 (51.37)	No difference from before	175 (53.19)
	2–3 times per week	20 (6.08)	A bit more	56 (17.02)	A bit more	25 (7.60)
	4 or more times per week	8 (2.43)	Much more	30 (9.12)	Much more	7 (2.13)
	Missing	36 (10.94)	Missing	45 (13.68)	Missing	45 (13.68)
Had sex in exchange for money, material goods, favors, drugs, or shelter	Never	291 (88.45)	Much less	7 (2.13)	Much less	5 (1.52)
	Once a month or less	2 (0.61)	A bit less	0	A bit less	0
	2–4 times per month	1 (0.30)	No difference from before	264 (80.24)	No difference from before	266 (80.85)
	2–3 times per week	0	A bit more	0	A bit more	1 (0.30)
	4 or more times per week	1 (0.30)	Much more	1 (0.30)	Much more	0
	Missing	34 (10.33)	Missing	57 (17.33)	Missing	57 (17.33)
Watched sexually explicit videos (pornography)	Never	73 (22.19)	Much less	27 (8.21)	Much less	36 (10.94)
	Once a month or less	86 (26.14)	A bit less	22 (6.69)	A bit less	44 (13.37)
	2–4 times per month	64 (19.45)	No difference from before	159 (48.33)	No difference from before	176 (53.50)
	2–3 times per week	50 (15.20)	A bit more	69 (20.97)	A bit more	30 (9.12)
	4 or more times per week	27 (8.21)	Much more	21 (6.38)	Much more	12 (3.65)
	Missing	29 (8.81)	Missing	31 (9.42)	Missing	31 (9.42)
Performed/watched sexual acts on a webcam	Never	268 (81.46)	Much less	11 (3.34)	Much less	19 (5.78)
	Once a month or less	21 (6.38)	A bit less	2 (0.61)	A bit less	6 (1.82)
	2–4 times per month	7 (2.13)	No difference from before	241 (73.25)	No difference from before	244 (74.16)
	2–3 times per week	1 (0.30)	A bit more	17 (5.17)	A bit more	5 (1.52)
	4 or more times per week	2 (0.61)	Much more	8 (2.43)	Much more	4 (1.22)
	Missing	30 (9.12)	Missing	50 (15.20)	Missing	51 (15.50)

**Table 5 ijerph-21-00096-t005:** Experience of sexual violence before, during, and after lockdown.

Violence-Related Variables		Pre-COVID-19 (about 3 Months before)	During COVID-19 Lockdown	After COVID-19 Lockdown	F _(1,312)_	Sign
How vulnerable did you feel for sexual harassment or sexual, physical, or emotional assault by someone who does not live in your house?	Not vulnerable	194 (58.97)	250 (75.99)	-	37.574	<0.001
	Slightly vulnerable	46 (13.98)	15 (4.65)	-		
	Neutral	25 (7.60)	21 (6.38)	-		
	A bit vulnerable	28 (8.51)	8 (2.43)	-		
	Very vulnerable	5 (1.52)	4 (1.22)	-		
	Missing	31 (9.42)	31 (9.42)	-		
Has a partner tried to restrict (online or phone) contact with your family?	No	243 (73.86)	239 (72.64)	244 (74.16)	1.516	0.219
	Yes, once	1 (0.30)	2 (0.61)	1 (0.30)		
	Yes, more than once	3 (0.91)	1 (0.30)	1 (0.30)		
	Not applicable to my situation	53 (16.11)	55 (16.72)	51 (15.50)		
	Missing	29 (8.81)	32 (9.73)	32 (9.73)		
Has a partner insulted you or made you feel bad about yourself?	No	206 (62.61)	206 (62.61)	212 (64.44)	2.156	0.143
	Yes, once	38 (11.55)	28 (8.51)	30 (9.12)		
	Yes, more than once	14 (4.26)	20 (6.08)	17 (5.17)		
	Not applicable to my situation	39 (11.86)	41 (12.46)	38 (11.55)		
	Missing	32 (9.73)	34 (10.33)	32 (9.73)		
Has a partner ever not provided money to run the house or look after the children, but has money for other things?	No	131 (39.82)	127 (38.60)	129 (39.21)	0.106	0.745
	Yes, once	1 (0.30)	4 (1.22)	1 (0.30)		
	Yes, more than once	5 (1.52)	1 (0.30)	2 (0.61)		
	Not applicable to my situation	141 (42.86)	142 (43.16)	143 (43.47)		
	Missing	51 (15.50)	55 (16.72)	54 (16.41)		
Has a partner slapped, pushed, hit, kicked, or choked you or thrown something at you that could hurt you?	No	239 (72.64)	218 (66.26)	236 (71.73)	0.199	0.656
	Yes, once	4 (1.22)	3 (0.91)	5 (1.52)		
	Yes, more than once	1 (0.30)	1 (0.30)	0		
	Not applicable to my situation	47 (14.29)	68 (20.67)	50 (15.20)		
	Missing	38 (11.55)	39 (11.85)	38 (11.55)		
Has a partner physically forced you to have sexual intercourse when you did not want to?	No	240 (72.95)	222 (67.48)	242 (73.56)	0	1
	Yes, once	3 (0.91)	4 (1.22)	3 (0.91)		
	Yes, more than once	0	0	1 (0.30)		
	Not applicable to my situation	43 (13.07)	61 (18.54)	40 (12.16)		
	Missing	43 (13.07)	42 (12.77)	43 (13.07)		
Has a partner made you have sexual intercourse when you did not want to because you were afraid of what your partner might do?	No	245 (74.47)	218 (66.26)	241 (73.25)	1.604	0.207
	Yes, once	7 (2.13)	2 (0.61)	5 (1.52)		
	Yes, more than once	1 (0.30)	1 (0.30)	0		
	Not applicable to my situation	35 (10.64)	63 (19.15)	40 (12.16)		
	Missing	41 (12.46)	45 (13.68)	43 (13.07)		

## Data Availability

Data are available upon reasonable request.

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
