# Peer review of "Changes in Sexual Behavior and Satisfaction and Violent Behavior during COVID-19 Lockdown: Explorative Results from the Italian Cross-Sectional Study of the I-SHARE Multi-Country Project"

_ijerph, 2024, doi:10.3390/ijerph21010096_

Round 1
Reviewer 1 Report
Comments and Suggestions for Authors
The study deals with an interesting topic, namely the sexual well-being of people during, pre and post the pandemic. This aspect has been quite neglected in research as well as in the field of mental and physical health, and it is important to find data on this aspect in the literature.
Some suggestions to improve the work that I think overall is appropriate for publication in this journal:
- When discussing violence against women during lockdown, I suggest including or adding more recent citations (e.g., Huldani et al., 2022; McNeil et al., 2023).
- Perhaps a reference to the use of pornography and sexual gratification in the relationship with the partner should be added (Irizzary et al., 2023).
- I think it is necessary to expand the literature cited, enrich the introduction, discussion and comparison with previous data in the literature and try to include more recent citations and scientific papers (example: Quaderi et al., 2023; Mauriki et al., 2022)
- Include a description of the measures that Italy has taken to combat the pandemic and make reference to the possible impact that these measures have had on mental well-being in general (Longobardi et al., 2019)
- For the research objectives, explain the hypotheses, if they have been formulated. Alternatively, you can state in the title that this is an exploratory study.
- The information provided in the previous points should be reflected in the discussion, which should generally appear well organized and structured.
- The limits of the research need to be expanded and elaborated, and in particular the cross-sectional nature of your study needs to be emphasized. Clear directions for future research could be given, bearing in mind that similar situations could occur in the future (which we hope will not happen!) and that the pandemic could have effects that will last over a longer period of time.
- A section on the practical implications (intervention and prevention, social policy, etc.) is necessary in my opinion.
Recommended references (not mandatory)
Huldani, H., Kamal Abdelbasset, W., Abdalkareem Jasim, S., Suksatan, W., Turki Jalil, A., Thangavelu, L., ... & Karami, M. (2022). Intimate partner violence against pregnant women during the COVID-19 pandemic: A systematic review and meta-analysis. Women & Health, 62(6), 556-564.
McNeil, A., Hicks, L., Yalcinoz-Ucan, B., & Browne, D. T. (2023). Prevalence and correlates of intimate partner violence during COVID-19: A rapid review. Journal of Family Violence, 38(2), 241-261.
Irizarry, R., Gallaher, H., Samuel, S., Soares, J., & Villela, J. (2023). How the Rise of Problematic Pornography Consumption and the COVID-19 Pandemic Has Led to a Decrease in Physical Sexual Interactions and Relationships and an Increase in Addictive Behaviors and Cluster B Personality Traits: A Meta-Analysis. Cureus, 15(6).
Qaderi, K., Yazdkhasti, M., Zangeneh, S., Behbahani, B. M., Kalhor, M., Shamsabadi, A., ... & Rasoal, D. (2023). Changes in sexual activities, function and satisfaction during the COVID-19 pandemic era: a systematic review and meta-analysis. Sexual medicine, 11(2), qfad005.
Mourikis, I., Kokka, I., Koumantarou-Malisiova, E., Kontoangelos, K., Konstantakopoulos, G., & Papageorgiou, C. (2022). Exploring adult sexual wellbeing and behavior during the COVID-19 pandemic. A systematic review and meta-analysis. Frontiers in Psychiatry, 13, 949077.
Longobardi, C., Morese, R., & Fabris, M. A. (2020). COVID-19 emergency: social distancing and social exclusion as risks for suicidal ideation and attempts in adolescents. Frontiers in psychology, 11, 551113.
Author Response
Reviewer 1
The study deals with an interesting topic, namely the sexual well-being of people during, pre and post the pandemic. This aspect has been quite neglected in research as well as in the field of mental and physical health, and it is important to find data on this aspect in the literature.
Some suggestions to improve the work that I think overall is appropriate for publication in this journal:
- When discussing violence against women during lockdown, I suggest including or adding more recent citations (e.g., Huldani et al., 2022; McNeil et al., 2023).
- Perhaps a reference to the use of pornography and sexual gratification in the relationship with the partner should be added (Irizzary et al., 2023).
- I think it is necessary to expand the literature cited, enrich the introduction, discussion and comparison with previous data in the literature and try to include more recent citations and scientific papers (example: Quaderi et al., 2023; Mauriki et al., 2022)
Response: Thank you for the suggested references. We have included some of the suggestions in the introduction and discussion of the paper.
- Include a description of the measures that Italy has taken to combat the pandemic and make reference to the possible impact that these measures have had on mental well-being in general (Longobardi et al., 2020).
Response: Thank you for this comment. We have added the information suggested by the reviewer as follows: “Restrictive measures consisted of maintaining interpersonal distance of at least 1 meter in social contacts, hand sanitization at the entrance of any place open to the public, suspension of educational activities in all schools and universities, closure of all un-necessary activities, specific restrictions for other public places such as offices and hospitals. Being in quarantine has been linked to a decline in mental well-being and the emergence of psychological issues like depression, anxiety, insomnia, and post-traumatic symptoms (Longobardi et al., 2020). The loss of routine and limited physical and social interactions may intensify feelings of isolation and loneliness, leading to psychological distress (Brooks et al., 2020). While existing research primarily focuses on the psychological well-being of adults and healthcare professionals, there is a notable gap in understanding the impact on sexual and reproductive health.”
- For the research objectives, explain the hypotheses, if they have been formulated. Alternatively, you can state in the title that this is an exploratory study.
Response: Thank you for this comment. We have changed the title as suggested since no previous hypothesis were made.
- The information provided in the previous points should be reflected in the discussion, which should generally appear well organized and structured.
Response: thank you for this comment. We have reviewed the discussion also considering the previous points raised by the reviewer.
- The limits of the research need to be expanded and elaborated, and in particular the cross-sectional nature of your study needs to be emphasized. Clear directions for future research could be given, bearing in mind that similar situations could occur in the future (which we hope will not happen!) and that the pandemic could have effects that will last over a longer period of time.
Response: Thank you for this comment. We have updated the section adding your indications: “The current study acknowledges certain limitations. Firstly, participants were recruited using a "snowball" technique and social media advertising, making it challenging to generalize the findings to the Italian population, despite the diverse range of participants. Secondly, the research design of this study is cross sectional although it asks about the perception of participants in 3 different time (pre-, during, and post- lockdown). The I-SHARE measure was specifically developed and adapted for the Ital-ian language in early 2020. While initial psychometric analyses have been undertaken, further research is needed to explore the validity and reliability of this measure. Lastly, the study relied on self-reported measures, potentially allowing respondents to falsify their responses. Another potential bias is that it was not possible to control whether participants have traveled during restriction measures in other countries, although February-June 2020 was not a period when people could travel freely. Future studies should consider how sexual and reproductive health has evolved not only since the emergency restriction measures, but especially in the long run, after the WHO's declaration of the end of the pandemic in June 2023. This temporal overview can be extremely useful in understanding the complexity of the pandemic effect and anticipating preventive sexual health measures for possible future emergencies.”
- A section on the practical implications (intervention and prevention, social policy, etc.) is necessary in my opinion.
Response: Thank you for your comment. We have modified the conclusion adding some practical implication of the study as follows “This study adds to the existing evidence of sexual behavior changes during the pandemic. There is now a lot of data about the impact of lockdown on sexuality, which has on one hand worsened sexual health problems and on the other hand highlighted new possibilities of engaging in one’s sexual life, such as self-eroticism and digital sex. These changes should be taken into consideration when planning responses to health emergencies and epidemics. This study can specifically provide some intervention and prevention directions related to social policy and clinical practice. For example, it is suggested by this study how in times of stress, sexuality turns out to be one of the strongly impacted elements. Because sexuality also plays an important role in relationships and stress relief, it is central to develop awareness campaigns that enhance the role of sexual health and increase access to specialized services that can indicate, clarify, and intervene where needed. Sexuality is certainly not the central element when it comes to airborne infectious pathogen emergencies, but it can be a resource for what concerns social restraint measures and a person's psychological and emotional health.”
Recommended references (not mandatory)
Huldani, H., Kamal Abdelbasset, W., Abdalkareem Jasim, S., Suksatan, W., Turki Jalil, A., Thangavelu, L., ... & Karami, M. (2022). Intimate partner violence against pregnant women during the COVID-19 pandemic: A systematic review and meta-analysis. Women & Health, 62(6), 556-564.
McNeil, A., Hicks, L., Yalcinoz-Ucan, B., & Browne, D. T. (2023). Prevalence and correlates of intimate partner violence during COVID-19: A rapid review. Journal of Family Violence, 38(2), 241-261.
Irizarry, R., Gallaher, H., Samuel, S., Soares, J., & Villela, J. (2023). How the Rise of Problematic Pornography Consumption and the COVID-19 Pandemic Has Led to a Decrease in Physical Sexual Interactions and Relationships and an Increase in Addictive Behaviors and Cluster B Personality Traits: A Meta-Analysis. Cureus, 15(6).
Qaderi, K., Yazdkhasti, M., Zangeneh, S., Behbahani, B. M., Kalhor, M., Shamsabadi, A., ... & Rasoal, D. (2023). Changes in sexual activities, function and satisfaction during the COVID-19 pandemic era: a systematic review and meta-analysis. Sexual medicine, 11(2), qfad005.
Mourikis, I., Kokka, I., Koumantarou-Malisiova, E., Kontoangelos, K., Konstantakopoulos, G., & Papageorgiou, C. (2022). Exploring adult sexual wellbeing and behavior during the COVID-19 pandemic. A systematic review and meta-analysis. Frontiers in Psychiatry, 13, 949077.
Longobardi, C., Morese, R., & Fabris, M. A. (2020). COVID-19 emergency: social distancing and social exclusion as risks for suicidal ideation and attempts in adolescents. Frontiers in psychology, 11, 551113.

Reviewer 2 Report
Comments and Suggestions for Authors
Thank you for the opportunity to review the manuscript titled "Changes in sexual behavior, satisfaction, and violent behavior during COVID-19 lockdown: results from the Italian cross-sectional study of the I-SHARE multi-country project" for consideration for publication. This study utilizes survey data to analyze how sexual behaviors may have been impacted by the COVID-19 pandemic. It utilizes a time-series model, where participants are asked about their experiences pre-pandemic, during the pandemic, and post-pandemic.
Overall, I think this study is scientifically sound, utilizes appropriate analysis techniques, and draws valid conclusions given the researchers' analyses. It is well-written and clearly presented. I have only three considerations for possible improvement:
First, the authors use a very broad definition of "sexual behavior" that includes cuddling, kissing, and holding hands. It seems to me that very few people would equate holding hands with sexual behavior, and rather might categorize such behaviors as "romantic behaviors." I would encourage the authors to consider either changing their title from the words "Changes in Sexual Behavior..." to "Change sin Romantic Behavior..." or to separate the terms to "Changes in Romantic and Sexual Behaviors..." Such notation would need to be altered throughout the manuscript as well.
Secondly, the authors report that their population targeted Italians and their perceptions/behaviors during the pandemic. however, recruitment of participants only required people living in Italy at the time of the survey. Was any data collected to identify if the participants were living in Italy prior to June 2020 through January 2021 when the survey took place? I recognize that this was not a time period where many people were travelling internationally, but even small variances within the data might affect analysis.
Lastly, given that the survey ended in January 2021, can the results really be considered post-pandemic? The pandemic itself, at least in the United States, was not declared "ended" until May 2023. Perhaps the authors would be better served by re-describing their results as "pre-pandemic, early pandemic, late-pandemic" rather than post-pandemic. I would also recommend a sentence or two reflecting this in the conclusions to bolster the need for follow-up, long-term studies.
Author Response
Reviewer 2
Thank you for the opportunity to review the manuscript titled "Changes in sexual behavior, satisfaction, and violent behavior during COVID-19 lockdown: results from the Italian cross-sectional study of the I-SHARE multi-country project" for consideration for publication. This study utilizes survey data to analyze how sexual behaviors may have been impacted by the COVID-19 pandemic. It utilizes a time-series model, where participants are asked about their experiences pre-pandemic, during the pandemic, and post-pandemic.
Overall, I think this study is scientifically sound, utilizes appropriate analysis techniques, and draws valid conclusions given the researchers' analyses. It is well-written and clearly presented. I have only three considerations for possible improvement:
First, the authors use a very broad definition of "sexual behavior" that includes cuddling, kissing, and holding hands. It seems to me that very few people would equate holding hands with sexual behavior, and rather might categorize such behaviors as "romantic behaviors." I would encourage the authors to consider either changing their title from the words "Changes in Sexual Behavior..." to "Change sin Romantic Behavior..." or to separate the terms to "Changes in Romantic and Sexual Behaviors..." Such notation would need to be altered throughout the manuscript as well.
Response: Thank you for this comment. We understand the comment and we partially agree with the reviewer. The definition of sexual behavior is wide and (based on the fonts and on different cultures) may include kissing and holding hands. Beside this, we do not feel it is appropriate to add the word romantic, as the focus of the paper is on sexuality anyway. Even by including this word (which if the reviewer demands we can add it in the title and text) the content of the paper would not change and would not appropriately focus on romantic behaviors.
Secondly, the authors report that their population targeted Italians and their perceptions/behaviors during the pandemic. however, recruitment of participants only required people living in Italy at the time of the survey. Was any data collected to identify if the participants were living in Italy prior to June 2020 through January 2021 when the survey took place? I recognize that this was not a time period where many people were travelling internationally, but even small variances within the data might affect analysis.
Response: Thank you for this comment. Since this was an international survey, the participant could choose language and country where he or she currently lived before participating to be directed to the most appropriate version of the survey. Of course, it is not possible to control whether people have traveled in the meantime despite, as the reviewer rightly states, February-June 2020 was not a period when people could travel freely. Even if subjects did travel, the number would be so small that it would not be able to significantly influence the results statistically. We added a sentence to that effect on the limits.
Lastly, given that the survey ended in January 2021, can the results really be considered post-pandemic? The pandemic itself, at least in the United States, was not declared "ended" until May 2023. Perhaps the authors would be better served by re-describing their results as "pre-pandemic, early pandemic, late-pandemic" rather than post-pandemic. I would also recommend a sentence or two reflecting this in the conclusions to bolster the need for follow-up, long-term studies.
Response: Thank you for this comment. We agree with the reviewer. We meant lockdown rather than pandemic. We have reworded in the text and included a comment on future studies as suggested.

Reviewer 3 Report
Comments and Suggestions for Authors
Introduction:
In general, the introduction section is well-written; however, it is challenging to understand/comprehend the context in which the paper is written. It still needs to justify (i.e., provide rationale) the Italian context in which the study is conducted mentioning the study’s significance. Therefore, the introduction section requires further strengthening to provide background on the significance of the current study's rationale. Additionally, in the introduction section, briefly provide background research on IPV in the Italian context to facilitate readers' understanding of the study's significance.
Materials and Methods:
Please specify who collected the data and elaborate on how data collection was administered. In the Materials and Methods section, create a separate heading to provide information on the variables used in the study. Explain how the author(s) measured and coded the variables. In its current form, the materials and methods section is incomplete.
Results:
The attributes can be reorganized in the text of Table 4, as it appears messy. Perhaps, categories can be reorganized within three categories, excluding the "missing." Alternatively, the current Table 4 can be placed in the appendix. The same can be done for Table 5. It seems unnecessary to include missing information in the analysis.
As the writing does not mention the variables and how they were coded as dependent and independent variables, I, as a reviewer, do not comprehend the Results section and the use of the Generalized Linear Model as a methodological tool. In its current form, the paper seems more like a "draft" and is not yet ready for peer review.
Discussion:
Sincerely, the paper needs to recode the variables and reanalyze them; the results may differ from the present ones. Then, the discussion could be based on the new results.
The inclusion and discussion of IPV are confusing. How was IPV measured? I could not figure out the coherence of the presentation of ideas in the manuscript. The inclusion of missing cases (some variables have 31%, 34%, 36%), such as "Engaged in sexual activities with casual partners" with 53% and "Used condoms/contraceptives when you had sex with casual partners" with 68.39% missing, is inappropriate. The current analysis is not suitable, and the results are not helpful for discussion in the current form.
Author Response
Reviewer 3
In general, the introduction section is well-written; however, it is challenging to understand/comprehend the context in which the paper is written. It still needs to justify (i.e., provide rationale) the Italian context in which the study is conducted mentioning the study’s significance. Therefore, the introduction section requires further strengthening to provide background on the significance of the current study's rationale. Additionally, in the introduction section, briefly provide background research on IPV in the Italian context to facilitate readers' understanding of the study's significance.
Response: Thank you for this comment. We have reviewed the introduction following your suggestion and the ones from the other reviewers. We hope that the current version could be clearer for the reader and more able to underline the significance of the study.
Materials and Methods:
Please specify who collected the data and elaborate on how data collection was administered. In the Materials and Methods section, create a separate heading to provide information on the variables used in the study. Explain how the author(s) measured and coded the variables. In its current form, the materials and methods section is incomplete.
Response: Thank you for this comment. To respect rules of the journal and be concise enough, all the main information related by the reviewer are reported in specific parts of the article. Authors took care of the collection and data elaboration for the current study. Some details (as reported on the text) on the I-SHARE consortium and the survey methods that were the same for all the countries involved, can be found in the study protocol (Michielsen et al., 2021). The instrument with coded variables can be found in the Supplementary materials. In any case we would like to emphasize that the questionnaire is not based on a psychometric measure but refers to questions constructed ad hoc by the consortium. Therefore, the coding is reported as such in the tables for each variable/item considered. As such, we feel that having an additional section specifying items already reported is unnecessary and redundant. in any case, following the reviewer's comment we have tried to be clearer on this point in the text.
Results:
The attributes can be reorganized in the text of Table 4, as it appears messy. Perhaps, categories can be reorganized within three categories, excluding the "missing." Alternatively, the current Table 4 can be placed in the appendix. The same can be done for Table 5. It seems unnecessary to include missing information in the analysis.
As the writing does not mention the variables and how they were coded as dependent and independent variables, I, as a reviewer, do not comprehend the Results section and the use of the Generalized Linear Model as a methodological tool. In its current form, the paper seems more like a "draft" and is not yet ready for peer review.
Response: Thank you for these comments. We have reviewed all the tables and adjusted look and contents. Related to missing responses, we strongly believe that they are important to be reported to be clearer to the reader the response rate and the significance of the results (as we stated in the text, most of the items were optional for the participants). We would like to specify to the reader that missing responses are never computed when running generalized linear models, which are based on mean scores of groups, so re-running the analysis makes no sense since the results will be the same. Results reported here are already related to mean scores which do not consider missing data in the analysis. In addition, we clarify that not reporting the number and percentage of missing response would be a serious omission of data. Related to the Generalized Linear Model, we have specified in the text dependent and independent variables in order to be clearer for the reader as suggested by the reviewer. We hope that the current version may be clearer for the reader and have cleared up the reviewer's doubts about the results.
Discussion:
Sincerely, the paper needs to recode the variables and reanalyze them; the results may differ from the present ones. Then, the discussion could be based on the new results.
The inclusion and discussion of IPV are confusing. How was IPV measured? I could not figure out the coherence of the presentation of ideas in the manuscript. The inclusion of missing cases (some variables have 31%, 34%, 36%), such as "Engaged in sexual activities with casual partners" with 53% and "Used condoms/contraceptives when you had sex with casual partners" with 68.39% missing, is inappropriate. The current analysis is not suitable, and the results are not helpful for discussion in the current form.
Response: Thank you also for these comments. As we specified in the previous point, there is no need to redo the analyses because the data from the generalized linear models are already cleaned of the missing. With respect to the comment about IPV, we agree with the reviewer that the use of the word IPV may be misleading with respect to the variables analyzed in the current study. We have slightly reworded the violence part of the discussions with a view to better clarify the results. Thank you again.

Round 2
Reviewer 1 Report
Comments and Suggestions for Authors.